# A Quantum Ant Colony Multi-Objective Routing Algorithm in WSN and Its Application in a Manufacturing Environment

**DOI:** 10.3390/s19153334

**Published:** 2019-07-29

**Authors:** Fei Li, Min Liu, Gaowei Xu

**Affiliations:** 1Department of Computer Science, Zhejiang University City College, Hangzhou 310015, China; 2College of Electronics and Information Engineering, Tongji University, Shanghai 201804, China

**Keywords:** wireless sensor network (WSN), energy, ant colony optimization (ACO), routing algorithm, quantum-inspired evolutionary algorithms

## Abstract

In many complex manufacturing environments, the running equipment must be monitored by Wireless Sensor Networks (WSNs), which not only requires WSNs to have long service lifetimes, but also to achieve rapid and high-quality transmission of equipment monitoring data to monitoring centers. Traditional routing algorithms in WSNs, such as Basic Ant-Based Routing (BABR) only require the single shortest path, and the BABR algorithm converges slowly, easily falling into a local optimum and leading to premature stagnation of the algorithm. A new WSN routing algorithm, named the Quantum Ant Colony Multi-Objective Routing (QACMOR) can be used for monitoring in such manufacturing environments by introducing quantum computation and a multi-objective fitness function into the routing research algorithm. Concretely, quantum bits are used to represent the node pheromone, and quantum gates are rotated to update the pheromone of the search path. The factors of energy consumption, transmission delay, and network load-balancing degree of the nodes in the search path act as fitness functions to determine the optimal path. Here, a simulation analysis and actual manufacturing environment verify the QACMOR’s improvement in performance.

## 1. Introduction

Recent years have seen a worldwide interest in Wireless Sensor Network (WSN) [1] technology, which has been considered one of the most promising technologies in smart manufacturing. Actually, the development tendency of WSN is in accordance with its context of Industry 4.0 [2]. Together with the Industrial Internet, the Internet of Things (IoT) [3], whose kernel is WSN, contributes to the achievement of the connectivity and communication of Cyber-Physical Systems (CPS) [4]. WSN techniques are appropriate for long-term data acquisition for IoT representation in an industrial environment. 

WSNs are distinguished from traditional wireless networks by their dissimilar purposes: WSNs are data-centric, while the latter aim for data transmission. In traditional wireless networks, such as Ad hoc and Wireless Local Area Networks (WLANs), the main task is to find the low-latency path between the source node and the destination node, and to improve the utilization of the whole network in order to avoid communication congestion and simultaneously balance network flow. However, in WSNs, a routing method has two main functions: to find the optimal path from the source node to the destination node, and to transmit a data packet along that path. The main aim of network routing improvement is to extend network life and prevent connection errors [5]. The routing method’s emphasis is on energy efficiency, because of limited node energy and long lifetime requirements. Meanwhile, since the number of sensor nodes tends to be very large, and these nodes can only obtain local topological information, a suitable route should be chosen by considering local network information. 

Since the network is resource- and power-limited, general wireless communication network routing methods are not well-suited for WSNs, especially in industrial fields in which there is demand for high performance in energy efficiency and longevity. Accordingly, some routing approaches have emerged, such as swarm intelligence-based schemes [5,6]. Social insect colonies, such as those of ants and honeybees [7,8], have complex collective behaviors and decentralized management structures, which are similar to parallel, dynamic, and distributed systems. Researchers have studied ant colony optimization (ACO)-based routing schemes to develop high-performance routing methods [9]. 

In order to improve the limitations of ACO-based routing methods, such as earlier stagnation and slow astringency, this paper considers the idea of using quantum-inspired evolutionary algorithms (QEAs) [10,11] and ACO together, balancing load, real-time transmission, and energy consumption with a multi-objective fitness function. A novel and efficient routing approach for WSNs, called the Quantum Ant Colony Multi-Objective Routing (QACMOR) algorithm, is proposed accordingly. In QACMOR, some quantum computing mechanisms of QEAs, including the quantum bit (qubit) and the quantum rotation gate, are introduced into ACO. The former represents the node’s pheromone, and the latter updates it. QEAs are able to avoid premature convergence with a simple implementation, which has more potential for solving large-scale problems than do other general evolutionary algorithms. In multiple objectives, more attention is paid to computation speed by using the look-up table of the rotation angle of QEAs and setting a time-delay factor in fitness function.

The rest of the paper is organized as follows: Section 2 presents the literature review on WSN routing methods. Section 3 sheds light on ACO-based routing in detail. Section 4 explains the proposed QACMOR approach. Section 5 shows the experimental results of performance evaluation and case study validated in a continuous steel casting production line. Finally, Section 6 discusses conclusions and future work.

## 2. Literature Review

The routing protocol of WSNs should be devised with properties such as energy efficiency, scalability, robustness, and rapid convergence, compared to that of traditional networks. A large number of routing methods have been proposed. Roughly, they can be divided into four categories through the analysis of relevant literature—that is, data-centric, clustering, geographic location-based, and Quality of Service (QoS)-based routing methods.

Data-centric routing was proposed to reduce the flooding overhead caused by transmitting query and data information. In data-centric routing, data request and collection are based on data attributes, rather than only using local interactions [12,13]. Clustering is the most common technique used for achieving energy-efficient and scalable performance in large-scale sensor networks. Cluster formation is a process whereby sensor nodes decide which cluster head they should associate with among multiple clusters [14,15]. The low-energy adaptive clustering hierarchy (LEACH) [15], a typical cluster-based algorithm, divides a sensor network into a set of clusters, through which energy consumption is balanced and reduced. In geographical routing, the physical location of the sensor node is used to guide the path that a packet takes in the network [16,17]. In some cases of WSN application, a higher-communication QoS is demanded, such as reliability and real-time data transmission. The method in [18,19] can be classified in this category.

Routing methods based on swarm intelligence have robust, adaptive, and scalable performance, suitable for autonomous distributed systems [20,21]. Inspired by the foraging principles of honeybees, Saleem et al. [22] proposed a distributed and decentralized routing protocol called the BeeSensor protocol. Camilo et al. [23] studied the application of the ACO metaheuristic to solve the routing problem in WSNs, and came up with an energy-efficient, ant-based routing algorithm (EEABR). Zungeru et al. [24] improved the EEABR algorithm by applying a new scheme to intelligently initialize and update routing tables, reducing the flooding ability of ants for congestion control. In [25], a self-adaptive routing mechanism is presented to ensure reliability and efficiency during data transmission by adopting the dissemination of a pheromone as a model for dealing with dynamic changes in WSN.

QEAs are based on the concept and principles of quantum computing, such as the quantum bit and the superposition of states. As a kind of evolutionary algorithm, a QEA is also characterized by the representation of the individual, the evaluation function, and population dynamics. Learning from the quantum rotation gate strategy of QEAs, Xing et al. [26] introduced an adaptive evolution mechanism for QoS multicasting in IP/DWDM networks, which allowed each chromosome in a population to update itself to a fitter position according to its own situation. 

## 3. Preliminaries

### 3.1. Energy Consumption Model 

Communication is the activity responsible for the bulk of the energy consumption in WSNs [27]. An energy consumption model used in Reference [27] is applied in this study (see Figure 1). 

Assumptions are that: the data can reach every node from its neighbors; the data contain information on distance and residual energy; the radio circuit in the sensor has a power control, and can expend the minimum required energy to reach the intended recipients; and radio circuit can be turned off to avoid receiving unintended transmissions. The transmission computation costs and receiving costs for a k-bit message at a certain distance *d* are shown as follows:

Transmitting
(1)$$E_{T}(k,d)=E_{elec}\times k+E_{amp}\times k\times d^{2}$$

Receiving
(2)$$E_{R}(k)=E_{elec}\times k+E_{BF}\times k$$

Total energy cost
(3)$$E=E_{T}+E_{R}$$where $E_{elec}=50nJ/bit$, $E_{amp}=100pJ/bit/m^{2}$ for the transmitter amplifier, and $E_{BF}=5nJ/bit$ when beamforming is used. *d* represents the distance of two nodes, and *k* represents the number of message bits.

Thus, by decreasing the communication distance and the volume of data to transmit, energy can be saved.

### 3.2. Basic Ant-Based Routing (BABR) Algorithm 

In ACO, ants exchange data by pheromones, and according to the positive feedback principle, a path with a high density of pheromones has a higher probability of being selected. Such optimization can be adapted to implement basic ant-based routing for WSNs [9,23]:

**Step 1**: At regular intervals, a forward ant *k* starts to move from the source node toward the destination. While moving, the identifiers of every visited node are recorded in a list, *M_k_*, and each forward ant avoids traversing a node that has been visited previously.

**Step 2**: At each node *r*, a forward ant selects the next hop node in accordance with a certain probability distribution:(4)$$P_{k}(r,s)=\left\{ \begin{array}{l} \frac{{\left[T(r,s)\right]}^{\mu }\cdot {\left[E(s)\right]}^{\nu }}{\sum_{s\notin M_{k}}{\left[T(r,s)\right]}^{\mu }\cdot {\left[E(s)\right]}^{\nu }},\;\mathrm{if}\;s\notin M_{k}\\ 0,\;\;\;\;otherwise \end{array}\right.$$where $P_{k}\left(r,s\right)$ is the probability of individual *k* that moves from node *r* to node *s*, and *T* is the routing table at each node with the amount of pheromone on the link (*r*, *s*) stored. *E* represents the heuristic information given by $1/\left(C-e_{s}\right)$ (*C* is the initial energy level of the nodes and *e_s_* is the actual energy level of node *s*), and *μ*, *ν* are weight parameters that signify the importance of pheromones versus heuristics. 

**Step 3**: When a forward ant reaches the destination, a backward ant goes back along the links that the forward ant has visited. Before moving, the amount of pheromones that the ant will drop during the trip is computed: (5)$$\Delta T_{k}=\frac{1}{N-Fd_{k}}$$where *N* is the total number of nodes, and *Fd_k_* is the distance traveled by the forward ant *k*.

**Step 4**: Whenever a node *r* receives a backward ant from a neighbor node, the routing table is updated:(6)$$T_{k}\left(r,s\right)=\left(1-\rho \right)T_{k}\left(r,s\right)+\Delta T_{k}$$where *ρ* is a coefficient, and then (1 − *ρ*) represents the evaporation of pheromones.

**Step 5**: Once a backward ant returns to the source node, the next interval is continued.

After several iterations, each node will find the best neighbors to which to send a data packet. While the ability and robustness of the ACO-based method qualify it to find a good solution, it still has the possibility of getting stuck in slow astringency and early stagnation.

## 4. The QACMOR Routing Method

This section first introduces the basic concepts and rules of QEAs, and then elaborates on the QACMOR algorithm for WSNs routing.

### 4.1. Mechanisms of QEAs

#### 4.1.1. Basic Elements of QEAs

The memory unit in a classical computer is the bit, which only has two states: “0” or “1”, whereas the smallest information unit in QEAs is defined as the qubit [10,11]. A qubit could be in the “0” state, the “1” state, or in a linear superposition of both, which is denoted as *α*|0〉 + *β*|1〉, where |0〉 and |1〉 represents the quantum state, and a pair of complex numbers (α,β) is defined with $|\alpha {|}^{2}+|\beta {|}^{2}=1$, and the value of $|\alpha {|}^{2}$ and $|\beta {|}^{2}$ indicates the probability of the “0” state and the “1” state, respectively.

A qubit with the size of *n* can be represented as the following, which has 2*^n^* kinds of states:(7)$$\left(\begin{array}{l} \alpha_{1}\\ \beta_{1} \end{array}|\begin{array}{l} \alpha_{2}\\ \beta_{2} \end{array}|\ldots \left|\begin{array}{c} \alpha_{i}\\ \beta_{i} \end{array}\right|\ldots |\begin{array}{l} \alpha_{n}\\ \beta_{n} \end{array}\right)$$

For example, a quantum individual with three qubits is given like this:(8)$$(\begin{array}{l} \frac{1}{\sqrt{2}}\;\\ \frac{1}{\sqrt{2}} \end{array}|\;\begin{array}{l} \frac{1}{\sqrt{2}}\;\\ \frac{-1}{\sqrt{2}} \end{array}|\;\begin{array}{l} \frac{1}{2}\\ \frac{\sqrt{3}}{2} \end{array})$$

It can also be represented as:(9)$$\frac{1}{4}|000\rangle +\frac{\sqrt{3}}{4}|001\rangle -\frac{1}{4}|010\rangle -\frac{\sqrt{3}}{4}|011\rangle +\frac{1}{4}|100\rangle +\frac{\sqrt{3}}{4}|101\rangle -\frac{1}{4}|110\rangle -\frac{\sqrt{3}}{4}|111\rangle$$which means that the probabilities of the states |000〉, |001〉, |010〉, |011〉, |100〉, |101〉, |110〉, and |111〉 are 1/16, 3/16, 1/16, 3/16, 1/16wh, 3/16, and 1/16, separately.

Commonly, ξ(ξ⊂(−π,π]) denotes the phase of the qubit, and the *i^th^* bit phase is $\xi_{i}=\arctan(\beta_{i}/\alpha_{i})$. The position of $\xi_{i}$ in coordination is given in Figure 2.

#### 4.1.2. The Updating of Qubit in QEAs

In QEAs, the quantum rotation gate updates the qubit. The following formula represents a qubit which rotates *θ_i_* degrees from the original vector, ${\left(\begin{array}{cc} \alpha_{i} & \beta_{i} \end{array}\right)}^{T}$ to ${\left(\begin{array}{cc} \alpha_{i}^{\prime } & \beta_{i}^{\prime } \end{array}\right)}^{T}$
(10)$$\left[\begin{array}{l} \alpha_{i}^{\prime }\\ \beta_{i}^{\prime } \end{array}\right]=\left[\begin{array}{cc} \cos(\theta_{i}) & -\sin(\theta_{i})\\ \sin(\theta_{i}) & \cos(\theta_{i}) \end{array}\right]\left[\begin{array}{l} \alpha_{i}\\ \beta_{i} \end{array}\right]$$

*θ_i_* is the rotation degree according to the following formula:(11)$$\theta_{i}=\Delta \theta \times s(\alpha_{i},\beta_{i})$$
(12)$$\Delta \theta =5\times \exp\left(-t/t_{\max}\right)$$

In Formulas (11) and (12), Δ*θ* represents the rotation step, controlling the rotation speed; *t* represents the current number of iterations; and *t*_max_ represents the predefined maximal times of calculation determined by the scale of the problem. The function $s(\alpha_{i},\beta_{i})$ defines the direction:(13)$$s(\alpha_{i},\beta_{i})=(d_{\mathrm{ibest}}/d_{\mathrm{inow}})(\xi_{\mathrm{ibest}}-\xi_{\mathrm{inow}})$$where
(14)$$\begin{array}{l} d_{\mathrm{inow}}=\beta_{\mathrm{inow}}/\alpha_{\mathrm{inow}}\\ d_{\mathrm{ibest}}=\beta_{\mathrm{ibest}}/\alpha_{\mathrm{ibest}}\\ \xi_{\mathrm{ibest}}=\arctan(\beta_{\mathrm{ibest}}/\alpha_{\mathrm{ibest}})\\ \xi_{\mathrm{inow}}=\arctan(\beta_{\mathrm{inow}}/\alpha_{\mathrm{inow}}) \end{array}$$

In Formula (14), $\alpha_{\mathrm{inow}}$, $\beta_{\mathrm{inow}}$, $\alpha_{\mathrm{ibest}}$, $\beta_{\mathrm{ibest}}$ are the probability of the *i^th^* qubit of the current and optimal solution, respectively. Finally, if $s(\alpha_{i},\beta_{i})<0$, the *θ_i_* rotates clockwise—otherwise, it rotates counterclockwise. 

### 4.2. The QEAs in QACMOR 

#### 4.2.1. Representing the Pheromone with Qubit

In QACMOR, a qubit represents the pheromone for a population with the size of *m* individuals—that is, $Q=\left(q_{1},q_{2},\ldots ,q_{j},\ldots q_{m}\right)$, j=1,2,…,m, and
(15)$$q_{j}=\left(\begin{array}{l} \alpha_{1}\\ \beta_{1} \end{array}|\begin{array}{l} \alpha_{2}\\ \beta_{2} \end{array}|\ldots |\begin{array}{l} \alpha_{n}\\ \beta_{n} \end{array}\right)$$where *n* is the number of qubits, $|\alpha_{i}{|}^{2}+|\beta_{i}{|}^{2}=1$, i=1,2,…,n.

#### 4.2.2. Updating the Pheromone with Quantum Rotation Gate

In QACMOR, similarly to Formulas (10) and (11), the quantum rotation gate G acting on the *i^th^* bit of the *j^th^* individual $q_{j}$ of solution *Q* is described as follows:(16)$$\left[\begin{array}{l} \alpha_{ji}^{\prime }\\ \beta_{ji}^{\prime } \end{array}\right]=G\left[\begin{array}{l} \alpha_{ji}\\ \beta_{ji} \end{array}\right]$$
(17)$$G=\left(\begin{array}{cc} \cos\theta_{ji} & -\sin\theta_{ji}\\ \sin\theta_{ji} & \cos\theta_{ji} \end{array}\right)$$
(18)$$\theta_{ji}=\Delta \theta_{ji}\times s(\alpha_{ji},\beta_{ji})$$where i=1,2,…,n, ${\left(\alpha_{ji}^{\prime },\beta_{ji}^{\prime }\right)}^{T}$ represents the updated bit, $\theta_{ji}$ is the rotation angle, $\Delta \theta_{ji}$ signifies the magnitude of the rotation angle, and $s(\alpha_{ji},\beta_{ji})$ is a function of $\alpha_{ji}$ and $\beta_{ji}$, and controls the direction of rotation. For the computation speed, the look-up table was applied to compute the rotation angle as shown in Table 1, which includes all feasible solutions. f(⋅) denotes the fitness function as Formula (20); $x_{ji}$ and $b_{i}$ represent the *i^th^* bit of the *j^th^* individual of the current solution x and the best solution b, respectively. The schematic diagram in Figure 3 shows the rotation gate polar plot for a qubit individual.

Additionally, a conventional binary solution is significantly important for performance evaluation, and can be obtained by observing the qubits. For example, it is assumed that $x_{i}$ (*i* = 1, 2, …, *n*) is a certain bit of the binary individual x, then $\alpha_{i}$ of the qubit individual is compared with a random number *w* (0 < *w* < 1). If ${\left|\alpha_{i}\right|}^{2}>w$, then set the value of $x_{i}$ to be “0”, otherwise set the value of $x_{i}$ to be “1”. Therefore, for $Q=\left(q_{1},q_{2},\ldots ,q_{j},\ldots q_{m}\right)$, j=1,2,…,m, its binary solution is $P=\left(p_{1},p_{2},\ldots ,p_{j},\ldots p_{m}\right)$, while $p_{j}(j=1,2,\ldots ,m)$ is a n-length binary individual, and then every element of $p_{j}$ (for example, $p_{ji}$) is determined by comparing $\alpha_{ji}$ of $q_{j}$ with *w*, 0 < *w* < 1.

### 4.3. The QACMOR Algorithm

The flowchart of the proposed approach is shown in Figure 4. The basic algorithm of QACMOR can be described as follows:

**Step 1:** The initialization step. Add every node and its neighbor nodes into the routing table. A forward ant is generated from source nodes which carry the information of source nodes, sink nodes, and passing nodes. The population is represented as $Q(t)=(q_{1}^{t},q_{2}^{t},\cdots ,q_{j}^{t},\cdots ,q_{m}^{t})$ with the size of *m* individuals, where $q_{j}^{t}(j=1,2,\cdots ,m)$ is the *j^th^* individual in the *t^th^* iteration. The representation is shown as:(19)$$q_{j}^{t}=\left(\begin{array}{l} \alpha_{j1}^{t}\\ \beta_{j1}^{t} \end{array}|\cdots \begin{array}{l} \alpha_{ji}^{t}\\ \beta_{ji}^{t} \end{array}|\cdots |\begin{array}{l} \alpha_{jn}^{t}\\ \alpha_{jn}^{t} \end{array}\right)$$where *n* is the number of qubits. Initialize $\alpha_{ji}$, $\beta_{ji}$(i=1,2,⋯,n) with $1/\sqrt{2}$. The maximum iterations are represented as *t_max_*, and the initial value of the current iterations *t* is 0.

**Step 2:** Compute the binary solution *P*(*t*). $P(t)=(p_{1}^{t},p_{2}^{t},\cdots ,p_{j}^{t},\cdots ,p_{m}^{t})$, $p_{j}^{t}(j=1,2,\cdots ,m)$ is a binary individual with n-length. The probable solution is obtained by measurement of *Q*(*t*). The value of element $p_{ji}$ in $p_{j}$ is determined by comparing $\alpha_{ji}$ of $q_{j}$ with *w*, 0 < *w* < 1.

**Step 3:** Generate the routing path. Assign *m* individuals into the source nodes at random. We used the state transition rule to generate the routing path of these individuals. In each step of the decision, an individual positioned on node *r* moves to the node *s* in line with Equations (4)–(6). 

**Step 4:** Evaluate the solution and store the best solutions in *B*(*t*). The evaluation function of the routing tree is shown as follows:(20)$$f(t)=\frac{1}{{\left[Z_{1}(t)\right]}^{C_{1}}{\left[Z_{2}(t)\right]}^{C_{2}}{\left[E_{r}(t)\right]}^{C_{3}}{\left[\sigma_{r}(t)\right]}^{C_{4}}}$$
(21)$$Z_{1}(t)=\sum Kd_{rs}^{\lambda },(r,s)\in Tree(t)$$
(22)$$Z_{2}(t)=\max Fd_{k}(t)$$
(23)F(t)=maxf(n),(n=0,1,…,t)where *C*_1_, *C*_2_, *C*_3_, and *C*_4_ are weight parameters, and $E_{r}(t)$ and $\sigma_{r}(t)$ are factors which describe the network load balance, and respectively represent the average value and standard deviation of the load for node *r*. *Z*_1_(*t*) is the energy consumption factor, *K* is an array which indicates the total number of leaf nodes extended from each node in the routing tree, λ is a parameter with a value from 2 to 4 which generally approaches 4, $d_{rs}$ is the distance of link (*r*,*s*), *Tree*(*t*) denotes the routing tree, *Z*_2_(*t*) is the time-delay factor, and *Fd_k_* is the distance traveled by the forward ant *k*.

After the sink node receives forward ant packages, evaluate the solution by Equations (20)–(23), and then save in *B*(*t*).

**Step 5:** Update the pheromone according to the rules of the quantum rotation gate, after receiving back the ant.

**Step 6:** If the current iterations are less than the maximum iterations, return to Step 3.

It should be noted that QACMOR is an evolutionary algorithm rather than a quantum algorithm, in spite of the fact that the proposed approach is based on quantum computing mechanisms. In QACMOR, some problems in basic ACO can be tackled. The representation of qubit introduces the probability research method, making the balance between exploration and exploitation easier than the conventional ACO algorithm, and adjustment of the magnitude of the rotation angle can make convergence speeds faster. Exploring the unused nodes by using heuristic information, as Formula (4) shows, updating the local pheromone according to Formula (5) and (6) in Step 2, and updating the global pheromone with the quantum rotation gate will generate population diversity, preventing the algorithm from becoming trapped in local convergence or premature stagnation.

## 5. Experimental Results

### 5.1. Performance Evaluation

Routing is a crucial process to consider in WSNs when dealing with multiple performance metrics, since routing decisions can impact network lifetime, packet delivery rates, and end-to-end packet delays [29]. Different performance metrics can be used for comparing different routing algorithms in WSNs. The main metrics considered in this paper to validate the performance of the proposed algorithm are as follows:(1)General property, such as communication distance, energy consumption, and hops.(2)Convergence rate, that is, the number of iterations needed to find an approximation to a fixed point.(3)Network lifetime, that is, the duration up to the time when data can no longer be forwarded due to the depletion of energy of the sensor nodes.

Sensor nodes are assigned at random. Figure 5 shows an instance in which the network range was 1000 *m*^2^, and the total number of nodes was 50. Each link between a node and its accessible neighbors was denoted by a dotted line. Figure 5 shows the optimal path obtained by QACMOR, shown as a solid red line. Source nodes were numbered 16, 21, 22, 24, 30, 47, and 50, and the sink node was numbered 1. Notice that the value of *t_max_* should be greater than the number of iterations for the algorithm to converge.

Three groups of experiments were conducted on a MATLAB simulation platform. Table 2 lists the values of parameters used in this simulation. 

In the first experiment, a comparison of the value of *F*(*t*) in cases in which the number of nodes ranges from 10 to 100 was conducted between two algorithms, that is, BABR and QACMOR. The curve lines in Figure 6 show that the values of *F*(*t*) for the two algorithms are same at the beginning, and descend as the number of nodes increases. Compared with BABR, the curve of QACMOR has a more sluggish downtrend. The reason for this is that QACMOR takes more properties into account, including energy efficiency, load balance, and time delay.

The aim of the second experiment was to estimate the convergence property by observing the optimal value *F*(*t*) of QACMOR and BABR when the number of iterations grows. As iterations grow, it can be seen in Figure 7 that the value of *F*(*t*) tends to be stable. In addition, we notice that QACMOR begins to converge at nearly 200 iterations, while it takes approximately 350 iterations for BABR. This demonstrates that QACMOR has a faster convergence rate than BABR.

The third experiment evaluates the network lifetime of QACMOR. The experiment is performed on the condition that the number of dead nodes grows. In Figure 8, the x-axis denotes the number of dead nodes, and the y-axis represents the lifetime of the network. It can be seen that the value of lifetime for QACMOR is consistently higher than the same value for BABR, the gap becomes bigger with increasing dead nodes, and maintains a fixed value after 35 nodes, nearly 900 *h*.

The above experimental results indicate that the QACMOR algorithm is capable of use as an efficient and reliable solution for routing, with balanced energy consumption and an improved network lifetime.

### 5.2. Case Study

In this section, a case study of a maintenance, repair, and overhaul (MRO) system for a steel manufacturing enterprise is illustrated, in order to evaluate the practicality of QACMOR. 

Some situations requiring WSNs, such as continuous steel casting lines, present unique characteristics, mainly due to their harsh industrial environment. In the case of a casting line, this is at high temperature and full of powder, dust, and noise. The installation site of the sensor nodes and sink makes it inconvenient to charge or replace the power supply. Therefore, network longevity should be considered. It is important to build routing algorithms which can be adapted to monitor equipment conditions and prolong the WSN lifetime as much as possible. Another major challenge in the harsh environment is insufficient QoS in WSNs, such as delay, bandwidth, and packet loss. 

The role of the online monitoring system is to obtain the status information of equipment, including temperature, pressure, and revolving speed. The system architecture is depicted in Figure 9. The complex structure of the continuous casting line made it difficult to install and deploy a reliable cable network, while a WSN had the ability to overcome the field wiring problem. In the WSN, these field data were sent to the Advanced RISC Machines (ARM)-based gateway for data collection, fusion, and processing. Then, the data were sent to the server. At the server, the collected data were imported into a database for further analysis and diagnosis of potential faults by the MRO system.

Sensor nodes distributed within the continuous casting line constitute the system’s perception layer. Figure 10 shows the installation site of three frame-offset wireless sensors on a segment. In this section, we chose one segment as the test object. Specifically, in one segment, 24 temperature sensors were used to collect information about the working status of the hydro-cylinders; 24 pressure sensors were installed to collect information about the bearings; eight revolving speed sensors were embedded to collect information about the rollers; and three frame-offset wireless sensors were installed onto each segment to monitor the displacement of the segment’s frame.

We conducted a test on one segment in a real casting-shop environment to compare the network lifetime of three algorithms—that is, BABR, AODV, and QACMOR, and verify the running practicality of QACMOR. In this test, the total number of nodes is 60, with one sink node and 59 sensor nodes for one segment. The parameter settings are listed in Table 3, which shows the same weight value (C_1_, C_2_, C_3_, C_4_) as that listed in Table 2 in Experiment 3 in Section 5.1. As in that experiment, the comparison of the whole network lifetime was made by observing the number of dead nodes. Results in Figure 11 indicate that in terms of time elapsed before first node death or total network lifetime, QACMOR still has an advantage over BABR, even in harsh working conditions.

## 6. Conclusions and Future Work 

ACO-based routing has been used widely in WSNs. To improve convergence performance and save energy consumption in basic ACO routing methods, quantum computing mechanisms were introduced in the QACMOR method. This paper studied two performance metrics: convergence rate and network lifetime, with reference to the features of industrial continuous steel casting production. Simulation results indicated that the algorithm proposed can rapidly obtain the optimal path with a fast convergence rate, and prolong the network lifetime. A WSN, based on the proposed QACMOR algorithm, was also deployed in an MRO system for a steel manufacturing enterprise. Physical WSN deployment and experiments showed that the proposed QACMOR algorithm is reliable in such applications, after consideration of packet loss based on our previous work [21,30]. In future work, focus and attention should be given to the potential synergies between WSNs and other existing and emerging technologies, such as Cloud Computing and Big Data, so as to improve their overall performance and efficiency.

## Figures and Tables

**Figure 1 sensors-19-03334-f001:**
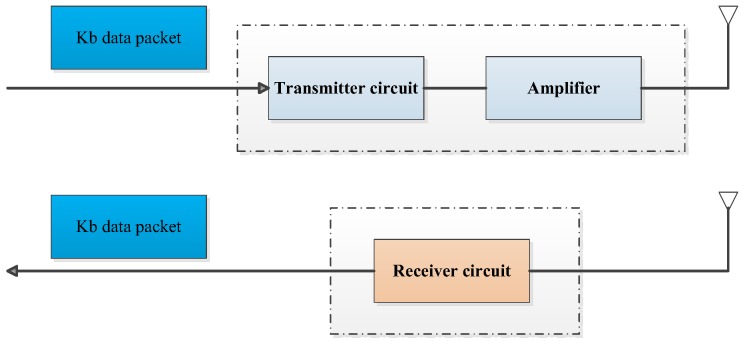
The energy consumption model.

**Figure 2 sensors-19-03334-f002:**
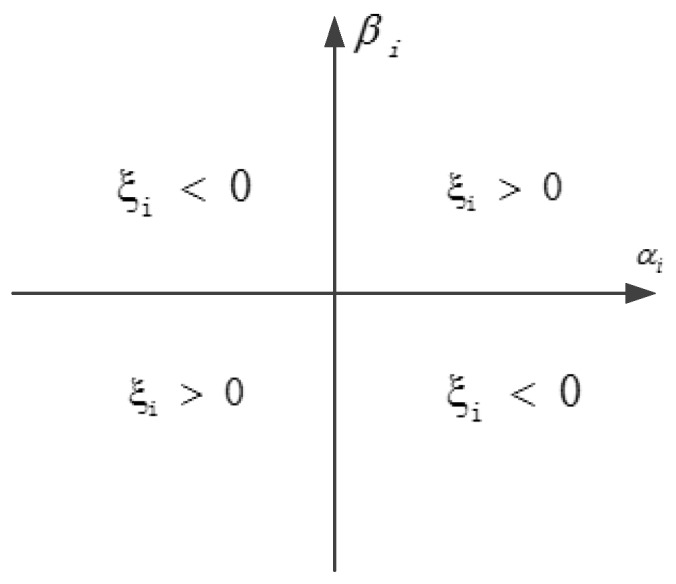
The position of *ξ_i_* in coordination.

**Figure 3 sensors-19-03334-f003:**
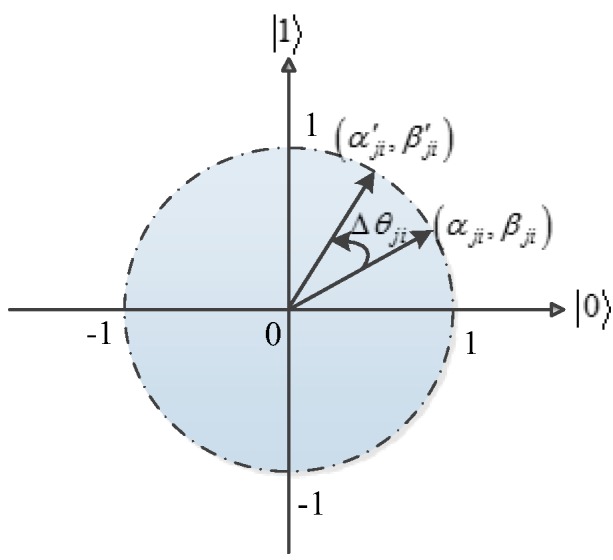
The polar plot of the rotation gate for a qubit individual.

**Figure 4 sensors-19-03334-f004:**
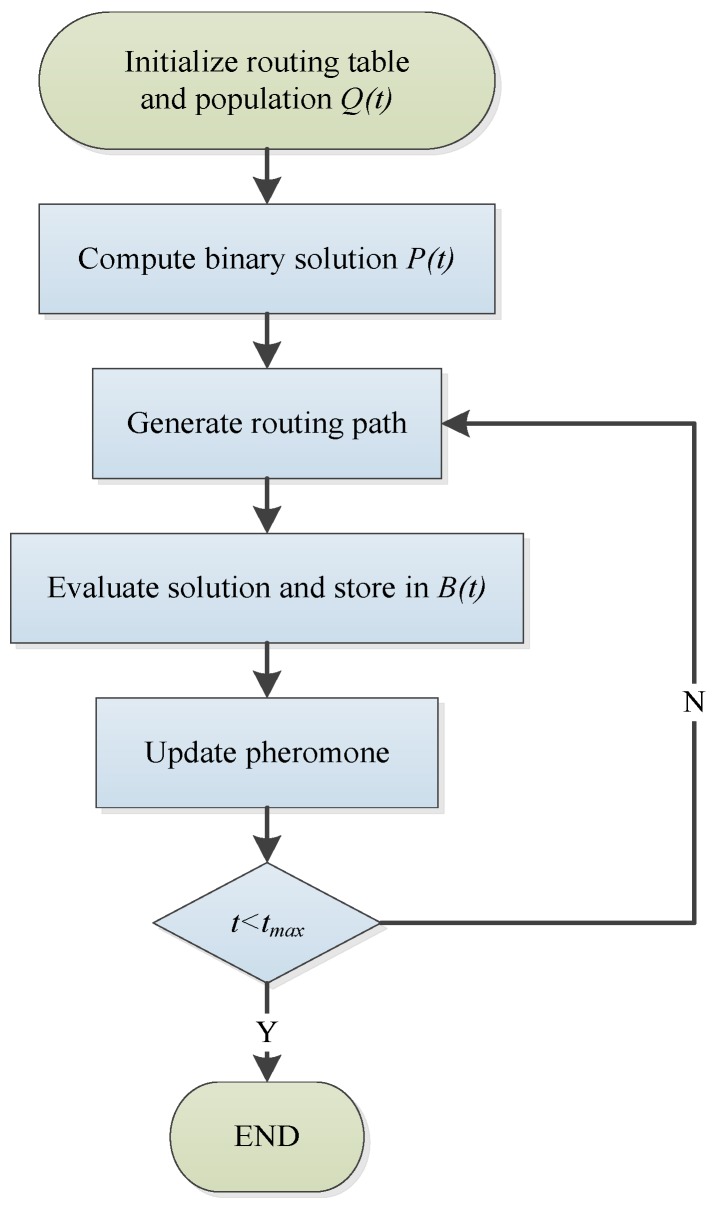
The process of the Quantum Ant Colony Multi-Objective Routing (QACMOR) approach.

**Figure 5 sensors-19-03334-f005:**
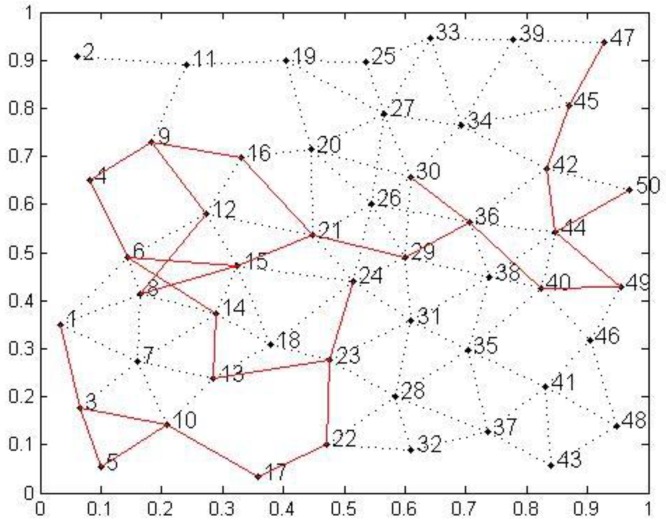
Optimal path in network topology.

**Figure 6 sensors-19-03334-f006:**
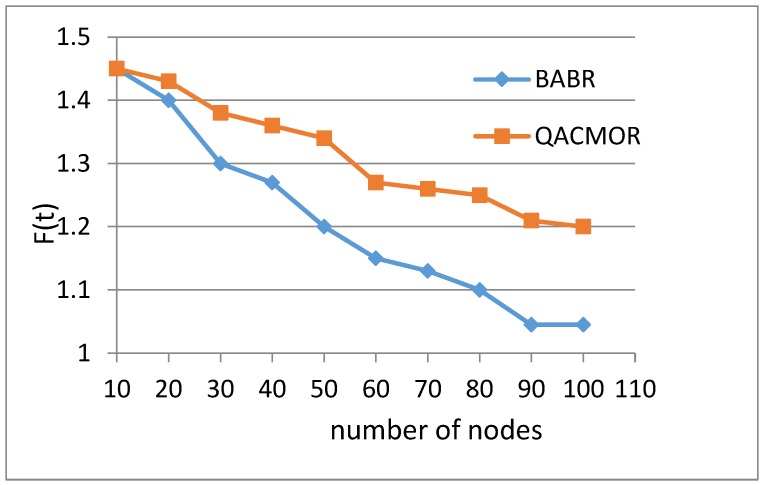
Value of optimal route vs. number of nodes.

**Figure 7 sensors-19-03334-f007:**
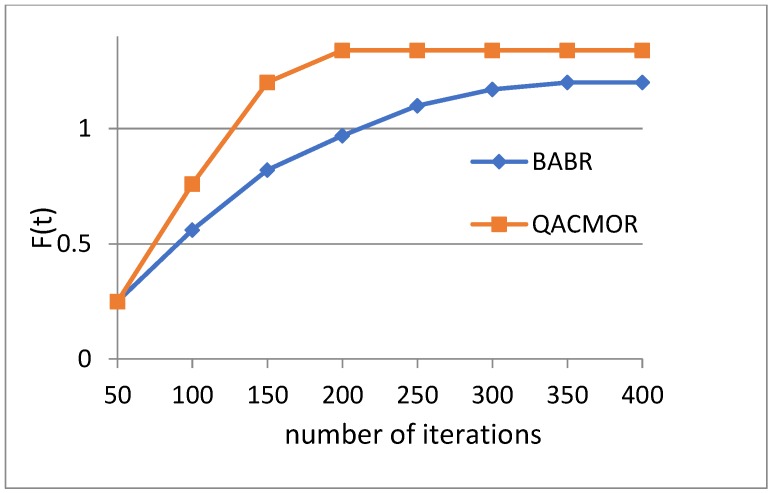
Value of optimal route vs. iterations.

**Figure 8 sensors-19-03334-f008:**
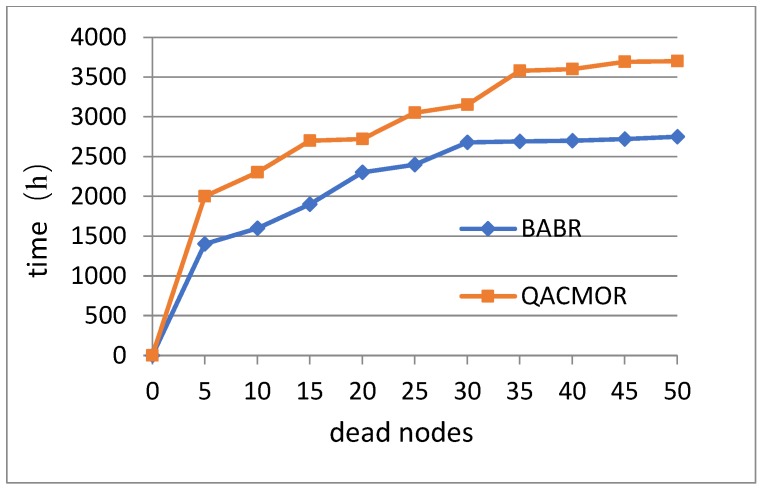
Time vs. number of dead nodes.

**Figure 9 sensors-19-03334-f009:**
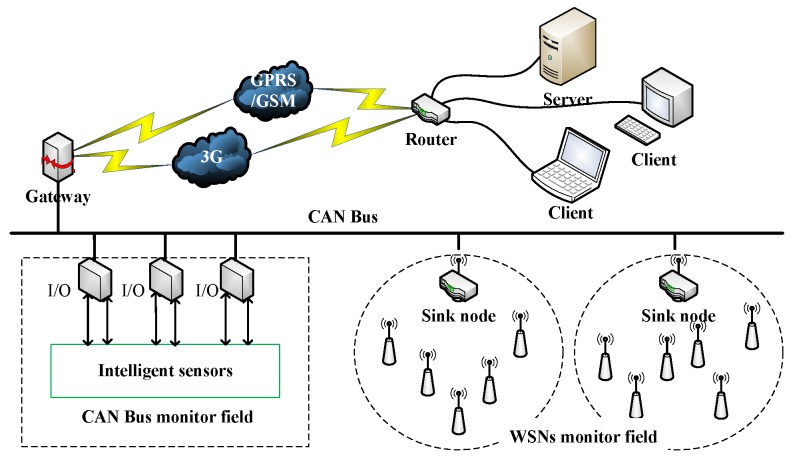
Online monitoring system architecture.

**Figure 10 sensors-19-03334-f010:**
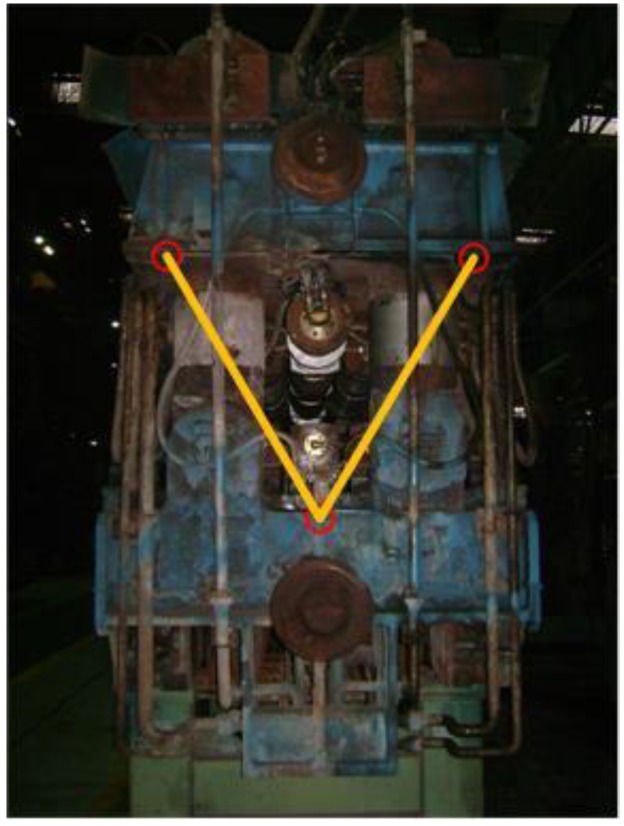
Diagram of three frame-offset wireless sensors.

**Figure 11 sensors-19-03334-f011:**
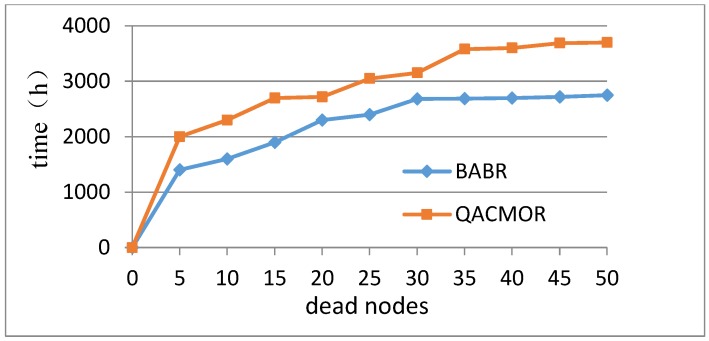
Comparison of network lifetimes in a continuous steel casting line.

**Table 1 sensors-19-03334-t001:** The look-up table of the quantum-inspired evolutionary algorithm (QEA) rotation angle [28].

$x_{i}$	$b_{i}$	f(x)>f(b)	$\Delta \theta_{i}$	$s\left(\alpha_{i},\beta_{i}\right)$			
				$\alpha_{i}\beta_{i}>0$	$\alpha_{i}\beta_{i}<0$	$\alpha_{i}=0$	$\beta_{i}=0$
0	0	False	0	0	0	0	0
0	0	True	0	0	0	0	0
0	1	False	0	0	0	0	0
0	1	True	0.05π	+1	−1	0	±1
1	0	False	0.01π	+1	−1	0	±1
1	0	True	0.025π	−1	+1	±1	0
1	1	False	0.005π	−1	+1	±1	0
1	1	True	0.025π	−1	+1	±1	0

**Table 2 sensors-19-03334-t002:** Parameter-setting in experiments.

Item	Experiment 1	Experiment 2	Experiment 3
Number of nodes	10, 20, 30, …, 100	50	50
Network range	1000 m^2^	1000 m^2^	5000 m^2^
Initial energy	/	/	0.5 J
*C* _1_	0.5	0.5	0.5
*C* _2_	0.1	0.1	0.1
*C* _3_	0.1	0.1	0.1
*C* _4_	0.1	0.1	0.1
*t_max_*	400	400	400

**Table 3 sensors-19-03334-t003:** Parameter-setting in case study.

Item	Value
Number of nodes	60
Network range	300 m × 280 m
Initial energy	0.5 J
*C* _1_	0.5
*C* _2_	0.1
*C* _3_	0.1
*C* _4_	0.1
*t_max_*	500

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
