# Peer review of "A Quantum Ant Colony Multi-Objective Routing Algorithm in WSN and Its Application in a Manufacturing Environment"

_sensors, 2019, doi:10.3390/s19153334_

Round 1

Reviewer 1 Report

The paper present a quantum algorithm based on ant colony multi-objective routing method with the objective to improve the performance.

In the quantum algorithm bit represents the pheromone. The improvement on performance is verified by both simulation and real real manufacturing application.

A section on quantum computing should be added. 

The quantum algorithm should be explained in a more detailed form.

Author Response

Comment 1:

The paper present a quantum algorithm based on ant colony multi-objective routing method with the objective to improve the performance

Response:

Thank you very much for your accurate conclusion. This article presents a quantum algorithm based on ant colony multi-objective routing method with the objective to improve the performance for monitoring the mechanical processing equipment.

Comment 2:

In the quantum algorithm bit represents the pheromone. The improvement on performance is verified by both simulation and real manufacturing application

Response:

Yes! The improvement on WSNs performance is verified by the simulation analysis and real application in a manufacturing environment.

Comment 3:

A section on quantum computing should be added.

Response:

Your suggestion is greatly appreciated. According to your suggestion, section 4.1.1 and 4.1.2 are added specially for representation of Q-bit and its updating.

Comment 3:

The quantum algorithm should be explained in a more detailed form.

Response:

Thank you very much for your advices. The section 4.1 has been divided into 4 sub-sections, which seperately introduce the Q-bit representation and Q-bit updating and its application in QACMOR. The new interpretation and analysis is more detailed

Reviewer 2 Report

- English could be improved, especially in the abstract. Some suggestions:

Abstract: Many complex manufacturing environments need to be monitored with Wireless Sensor Networks (WSNs), which demand both network life and data transmission quality. The traditional Ant Colony Optimization Algorithm (ACOA) does not fall easily into a local optimum and has slow convergence speed. Also, it has only a single performance objective: node energy consumption or transmission distance. A new WSN routing method named Quantum Ant Colony Multi-Objective Routing (QACMOR) can be used to monitor real manufacturing environments by representing pheromone with quantum bit, updating pheromone with quantum rotation gate, and introducing a multiple objective fitness function dependent on network load, energy consumption and transmission speed. The improvement on performance is verified by both simulation analysis and a real manufacturing application. 

- Figure 1, the energy consumption model, (lines 104, 105, 106) is simple and therefore does not have to be added, in my opinion.

- "108...The radios have power control and can expend the minimum required energy to reach the intended recipients. "

This assumption is not realistic. Although the authors show the improvement of their model over ACOA in Section 5.1, they do not justify the use of the LEACH energy model to compare with the network they implemented in the case study. The constants shown in line 115 (50 nJ/bit, 100 pJ/bit, 5 nJ/bit) are normally used in the LEACH protocol. However, the authors do not justify the use of these values to compare with the WSN they implemented for the case study. That is, are these values similar to the energy used by the nodes in the network they implemented in their case study?

-" 297 ... In a similar way to the experiment 3 inSubsection 5.1, the comparison of network lifetime was made by observing the number of dead nodes." 

It is not clear which experiment in Section 5.1 is experiment 3, presumably the one that has results presented in figure 7. The authors have not stated if the simulation experiment 3 had the same parameters - number of nodes, network size, etc. - as their case study network.

- Minor corrections could be made to the presentation of the results. For example, why does F(t) begin at -0.1 in Figure 6? The "R" in QACMOR is on a line below the rest of the acronym in some figures.

p.p1 {margin: 0.0px 0.0px 0.0px 0.0px; font: 9.0px 'Times New Roman'}

Author Response

Comment 1:

English could be improved, especially in the abstract.

Response:

Thanks for your suggestion. We read the manuscript many times and tried our best to polish the language, including grammar, expression, formula, table, and structure, especially in the abstract. We hope that the revisions meet with your approval.

Comment 2:

Figure 1, the energy consumption model, (lines 104, 105, 106) is simple and therefore does not have to be added, in my opinion.

Response:

Thank the reviewer for raising this issue. We agree that the energy consumption model is simple, but we adopted this simplified model to evaluate the performance of different algorithm.

Whether for ACOA or QACMOR, the first step of each iteration is to obtain information from their neighbors, including the distance between nodes, residual energy, hops to sink nodes, et, as described in Line 109 and 110 in revised manuscript. In the above two algorithms, the residual energy referred (denoted as es in Line 133 in revised manuscript) is calculated out according to this model.

So, this model should be kept for the algorithm integration.

Comment 3:

“The radios have power control and can expend the minimum required energy to reach the intended recipients. "

This assumption is not realistic. Although the authors show the improvement of their model over ACOA in Section 5.1, they do not justify the use of the LEACH energy model to compare with the network they implemented in the case study. The constants shown in line 115 (50 nJ/bit, 100 pJ/bit, 5 nJ/bit) are normally used in the LEACH protocol. However, the authors do not justify the use of these values to compare with the WSN they implemented for the case study. That is, are these values similar to the energy used by the nodes in the network they implemented in their case study?

Response:

Thank you very much for your careful review.

It should be affirmed that the first-order RF model cited from reference [27] should not be used solely for LEACH protocol. It is a common physical power consumption model for all wireless protocols including the transmitters and receivers.

In ideal case, no retransmissions or error correction is required in data layer. The first –order RF model can estimate the energy consumption of nodes in physical layer as the whole node energy consumption.

The power consumption of 50 nJ/bit, 100 pJ/bit, 5 nJ/bit mentioned in manuscript is slightly different from that of the actual product, which also be mentioned in reference [27], but it doesn’t affect the estimation a node energy consumption, which is positive proportional to transmission k bytes and square of transmission distance d2.

It can been seen that in reference [27] this model is also used to evaluate the power consumption in three different protocols: the power consumption of minimum power protocol, LEACH protocol and improved LEACH protocol.

To avoid misunderstanding, the word “LEACH” is deleted in Line 105 which is replaced with the word “in reference [27]”.

Comment 4:

" 297 ... In a similar way to the experiment 3 inSubsection 5.1, the comparison of network lifetime was made by observing the number of dead nodes." 

It is not clear which experiment in Section 5.1 is experiment 3, presumably the one that has results presented in figure 7. The authors have not stated if the simulation experiment 3 had the same parameters - number of nodes, network size, etc. - as their case study network.

Response:

Firstly, the experiment 3 is that one whose results are presented in figure 7 in original manuscript and figure 8 in revised manuscript. To make the order clear, a word “third’ is added in the Line 274. Secondly, the parameters in experiment 3 and case study have the same weight value C1, C2, C3, C4., but their number of nodes and network size are different, and the case study is in a worse working condition (as the word in Line 324).

To make clear all these meaning, the sentences between Line 320 and Line 324 have been rewritten like this:

The parameter settings are listed in Table 3, which shows the same weight value C1, C2, C3, C4 as listed in Table 2 in the experiment 3 in Subsection 5.1. In a similar way to that experiment, the comparison of whole network lifetime was made by observing the number of dead nodes. Results in Fig. 11 indicate that from the view of whether the first dead line time or the total network lifetime, QACMOR still has an advantage over ACOA even in a worse working conditions.

Comment 5:

Minor corrections could be made to the presentation of the results. For example, why does F(t) begin at -0.1 in Figure 6? The "R" in QACMOR is on a line below the rest of the acronym in some figures.

Response:

Thank you very much for your suggestion. We read the manuscript many times and many minor corrections have been made. We hope that the revisions meet with your approval.

Reviewer 3 Report

The paper is not easily readable as the English style is not very good; there are many mistakes and unclear sentences, even after the last revision.

Experimental results for the proposed algorithm, QACMOR, are compared to those of an algorithm (ACOA) that is mentioned in the abstract but is never explained along the paper (unless it is the basic ACO algorithm reported in Section 3.2, where, though, the ACOA acronym is not mentioned). Of course, this aspect should be improved and the reference algorithm should be explained with care, with the purpose of clarifying the differences between ACOA and QACMOR.

The presentation of Q-bits (“qubits” is a much more common term, but it is not a big problem) is inaccurate. The state of a quantum register can be expressed as in (7) only if the state can be factorized, and each Q-bit is in a “pure” state, but this is not true for the large majority of quantum register states, which involve some entanglement among the Q-bits. Put in other words, a register in the (7) form can also be expressed in the (9) form, but the inverse is not true, unless the state can be factorized. If the register can only be expressed as in (7) throughout the process, the power of quantum computation cannot be exploited. I read reference [10], which clarifies that the QEA algorithm must be executed on a classical (non-quantum) computer, and that QEA is a “quantum-inspired” algorithm, not a real quantum algorithm. This is not clear in this paper. The authors should present the quantum basics more correctly (probably, they should contact a physicist) and above all clarify what is the source of the performance improvement with respect to ACOA, given that there cannot be any speedup deriving from the use of a quantum computation device, even if such device were available. If this aspect is not explained, a reader that is not expert in quantum computing can incorrectly conclude that the performance improvement obtained with QACMOR relies on the properties of quantum computation.

Author Response

Response to Comments from Reviewer 3

Comment 1:

The paper is not easily readable as the English style is not very good; there are many mistakes and unclear sentences, even after the last revision.

Response:

Thanks for the comment. Considering even with several times revision of format and language of the paper by ourselves, there are still many mistakes and unclear sentences, we have invited a native English speaker who has co-revised with us and thoroughly improved the manuscript in terms of the English presentation. We hope the current version will meet up the requirements on English style.

Comment 2:

Experimental results for the proposed algorithm, QACMOR, are compared to those of an algorithm (ACOA) that is mentioned in the abstract but is never explained along the paper (unless it is the basic ACO algorithm reported in Section 3.2, where, though, the ACOA acronym is not mentioned). Of course, this aspect should be improved and the reference algorithm should be explained with care, with the purpose of clarifying the differences between ACOA and QACMOR

Response:

Thank you very much for your careful review.

It is a mistake we made due to our careless. ACO and ACOA indicates Ant Colony Optimization and Ant Colony Optimization Algorithm respectively. Although ACO acronym is a common term, and ACO or ACOA sometimes appears in literatures, ACOA is not a common term, which must be given an explanation of abbreviation.

But we deliberated and decided to replace the word “ACOA” with another more reasonable word “BABR” throughout this article. BABR indicates Basic ACO-Based Routing, which used in reference [9, 23]. In our article, we also introduced the BABR algorithm in section 3.2. To make the works in reference [9, 23] and our QACMOR more comparable, we think the acronym BABR is more appropriate.

Comment 3:

The presentation of Q-bits (“qubits” is a much more common term, but it is not a big problem) is inaccurate. The state of a quantum register can be expressed as in (7) only if the state can be factorized, and each Q-bit is in a “pure” state, but this is not true for the large majority of quantum register states, which involve some entanglement among the Q-bits. Put in other words, a register in the (7) form can also be expressed in the (9) form, but the inverse is not true, unless the state can be factorized. If the register can only be expressed as in (7) throughout the process, the power of quantum computation cannot be exploited. I read reference [10], which clarifies that the QEA algorithm must be executed on a classical (non-quantum) computer, and that QEA is a “quantum-inspired” algorithm, not a real quantum algorithm. This is not clear in this paper. The authors should present the quantum basics more correctly (probably, they should contact a physicist) and above all clarify what is the source of the performance improvement with respect to ACOA, given that there cannot be any speedup deriving from the use of a quantum computation device, even if such device were available. If this aspect is not explained, a reader that is not expert in quantum computing can incorrectly conclude that the performance improvement obtained with QACMOR relies on the properties of quantum computation.

Response:

We would like to thank the reviewer for the careful and thorough reading of this manuscript and related literatures and for the thoughtful comments and constructive suggestions.

Firstly, the word “Q-bit” has been replaced with the word “qubit” throughout the manuscripts.

Then, it must be stressed that the QACMOR or QACO (Quantum-inspired ACO) is an evolutionary algorithm rather than a quantum algorithm, in spite of the proposed approach is based on the quantum computing mechanisms.

Ant colony optimization is a kind of swarm intelligence algorithm. Similar to others algorithm such as particle swarm algorithm and fish swarm algorithm, they all simulate the foraging behavior of natural biological swarms. They use the phenomenon of simple individual behaviors but optimal group performance to solve many combinatorial optimization problems. Such as TSP (Travel Salesman Problem), knapsack problem, etc. In ACO, the behavior of ants has certain parallelism.

In the traditional ACO, there are two main defects: 1) the convergence speed is slow, mainly because the pheromone on each path is nearly equal at beginning of the ACO algorithm, the ant search every direction unordered, and 2) it is easy to fall into the local optimum. The main reason is that once an ant finds a suboptimal path, other ants also search along this suboptimal path. With the pheromone increase, the ant can't jump to the new path to find the best, which eventually leads to premature or stagnation. The above two defects have a sharp deterioration in performance when the number of nodes increases.

The QACO or QACMOR in this paper is a combination of ACO and QEA. They take advantage of the quantum coding and operators to overcome the above defects. In essence, AQCO or QACMOR is an improved ACO method, not a quantum computing. Of course, if quantum computing is available really, the feature of ant behavioral parallelism in ACO, and the research on QACO algorithm, can make the speed of QACO or QACMOR greatly improved.

In QACO or QACMOR, the individual in qubit representation has the advantage, as it is able to represent a linear superposition of states probabilities. Only one individual is enough to represent all the solutions, so the QACO can treat the balance between exploration and exploitation more easily than a conventional ACO algorithm by the probabilistic search. The advantage of convergence speed and overcoming the premature in QACO over the ACO has already been verified by analyze and experiment in traditional TSP or knapsack combination optimization problems.

Concretely, in step 5 of QACMOR, all ants use the quantum rotation gate to rotate the qubit to the optimal solution after the first iteration, which will increase the probability to find optimal path for all ants. Therefore, in the subsequent iteration, ants will search the optimal path under the dual effect of path pheromone and this direction optimized in previous iteration. However, in traditional ACO, ants can only make use of path pheromone. So the convergence speed of QACO or QACMOR is accelerated. In this process, the adjustment of rotation gate also take effect on the convergence speed. In QACO, ant exploring the unused nodes (Formula 4) and usage of rotation gate, or even non-gate in other literatures will increase the population diversity, and help ants jump from the local optimal solution.

Some words have been added into the end of section 4.3 to express the above content.

Round 2

Reviewer 3 Report

Though my previous comments have been addressed only partially, specifically the last one, the paper has been improved and I don't see much room for further improvement. In my opinion, the paper can be published.